# Adsorption Characteristics of Bacterial Cellulose Membranes Toward Methylene Blue Dye in Aqueous Environment

**DOI:** 10.3390/gels11090721

**Published:** 2025-09-10

**Authors:** Zimu Hu, Christopher R. Brewer, Austin J. Pyrch, Ziyu Wang, Dhanush U. Jamadgni, Wendy E. Krause, Lucian A. Lucia

**Affiliations:** 1Department of Textile Engineering, Chemistry and Science, North Carolina State University, Raleigh, NC 27606, USA; zhu10@ncsu.edu; 2Department of Chemistry, North Carolina State University, Raleigh, NC 27606, USA; christopher.brewer@utdallas.edu (C.R.B.);; 3Lampe Joint Department of Biomedical Engineering, The University of North Carolina at Chapel Hill and North Carolina State University, Raleigh, NC 27695, USA; zwang77@ncsu.edu; 4Comparative Medicine Institute, North Carolina State University, Raleigh, NC 27695, USA; 5Department of Materials Science and Engineering, North Carolina State University, Raleigh, NC 27606, USA; dudayas@ncsu.edu; 6Department of Forest Biomaterials, North Carolina State University, Raleigh, NC 27606, USA

**Keywords:** bacterial cellulose, adsorption, methylene blue, dye

## Abstract

Water pollution has escalated to critical levels in recent years as evident by the multiplicity of contaminants found in potable water sources. A point-source major contributor is the textile industry, which discharges substantial amounts of dye into rivers and lakes. Bacterial cellulose (BC), a renewable and low-cost nanocellulose material, has emerged as a potential solution addressing dye removal from these contaminated waters. Methylene Blue (MB) was selected as a representative dye for our adsorption studies. As a baseline for evaluating efficacy, BC was dried using three different methods: freeze-drying, oven-drying, and room-temperature drying. The adsorptive behavior of these dried BC samples toward MB in an aqueous environment was evaluated. Furthermore, to elucidate the structure–property relationship of dried BC, several characterization techniques were employed. Our studies revealed that freeze-dried BC exhibited the highest initial adsorption rate, while oven-dried BC demonstrated the overall highest adsorption capacity. Moreover, the adsorption data corresponded well with pseudo-second-order and Freundlich isotherm models. This investigation provides a comprehensive understanding of how BC, dried through different methods, performs in the adsorption of MB by establishing a baseline for future research.

## 1. Introduction

While industrial development has brought numerous conveniences, it has also resulted in significant pollution problems, including the contamination of water, air, and soil bioresources. These pollution issues pose serious threats to both human health and ecological systems [1]. The textile industry is one of the largest point-source contributors to water pollution, by releasing vast quantities of dyes and chemical by-products into receiving water bodies. Reports indicate that the textile industry contributes to about 75% of the global dyestuff production. This industry utilizes over 10,000 different dyes to color various textile products [2]. In recent years, the annual production of textile dyes has surpassed 700,000 tons [3]. These substances not only alter the aesthetic quality of water, but also interfere with photosynthesis in aquatic plants, disrupt aquatic life, and accumulate in the food chain, posing long-term health risks to humans and animals [1,4].

Various treatment methods have been developed and implemented. Traditional techniques, such as physico-chemical processes, oxidation, biological treatments, ion-exchange, and membrane filtration, have been widely used [5]. However, these conventional methods face significant challenges in simultaneously meeting multiple requirements, including high removal efficiency, non-secondary pollution, cost-effectiveness, and scalability. Consequently, the development of advanced methods for removing contaminants from aqueous solutions has become a crucial strategy for protecting the environment and human health [6]. Adsorption has gained attention for its efficiency and simplicity [7]. Materials such as activated carbon, zeolites, and emerging nanomaterials are being studied for their high adsorption capacities and ability to target specific dye molecules [8]. Additionally, innovative approaches such as the use of biosorbents have shown promise in dye removal applications. These materials offer environmentally friendly and sustainable solutions, with high surface areas, reusability, and biodegradability [9].

Bacterial cellulose (BC) is a type of nanocellulose that has gained popularity in wastewater treatment due to its high surface area, biocompatibility, and excellent tensile strength [10]. Produced through a bottom-up approach by microorganisms such as *Komagataeibacter xylinus*, BC synthesis involves the utilization of monosaccharides and nitrogen compounds [11,12]. The process starts with the synthesis of glucan chains in the inner membrane of bacteria, which are then secreted into nano-sized protofibrils that aggregate into BC fibers [13,14,15]. Compared with plant-derived cellulose hydrogels, BC hydrogels possess a unique nanofibrillar three-dimensional porous structure, higher purity and crystallinity, and can be directly produced by microbial fermentation without complex chemical treatments. These advantages have been further emphasized in recent studies, which highlighted the excellent reinforcing potential of BC owing to its high aspect ratio and biocompatibility [16]. Collectively, these intrinsic features provide the structural basis for the adsorption performance of the dried BC membranes investigated in this study [17,18]. Despite its advantages, producing BC using purified commercial mediums is costly and often results in pH alterations, longer incubation periods, and lower yields [19,20]. To address this, alternative low-cost resources such as food waste and fruit peels have been explored. In this study, BC is produced as a by-product of kombucha tea fermentation, utilizing a symbiotic culture of bacteria and yeast (SCOBY), offering an economical and sustainable approach to BC production.

BC hydrogels are inherently attractive as versatile and porous platforms for incorporating inorganic photoactive nanostructures, enabling the development of advanced hybrid and nanocomposite materials for water purification. However, the transition from hydrogel to dried membrane can markedly alter the material’s porosity, surface area, and adsorption behavior. Therefore, it is critical to evaluate how specific drying techniques affect the final properties of BC. Previous studies have mainly focused on BC produced by pure bacterial strains under well-defined culture conditions [13,21]. In contrast, much less attention has been given to BC derived from SCOBY cultivated in complex media, despite its increasing accessibility and potential scalability. In this case, we investigate SCOBY-derived BC membranes to systematically compare different drying methods, aiming to evaluate their ability to adsorb dyes such as methylene blue (MB) and to provide new insights into tailoring BC for effective environmental remediation applications.

Common drying methods for BC hydrogels include freeze-drying, oven-drying, room-temperature drying, and supercritical CO_2_ drying [13,21]. In this work, we selected the first three methods because they are relatively low-cost, simple, and straightforward to implement, aligning with our aim of exploring practical strategies to produce efficient dye adsorbents. The adsorptive behavior of the dried BC toward MB, a common textile dye, was then evaluated.

To elucidate the structure–property relationships of freeze-dried (FD), oven-dried (OD), and room-temperature dried (RD) BC, and to further understand how different drying methods affect the adsorptive behavior of BC, several characterization techniques were employed. Field emission scanning electron microscopy (FE-SEM), confocal laser scanning microscopy (CLSM), and atomic force microscopy (AFM) were utilized to examine the surface morphology of the BC samples. FE-SEM and transmission electron microscopy (TEM) were used to characterize the cross sections of the BC samples. X-ray microscopy (XRM) provided insights into the 3D structure of the BC samples. Differential scanning calorimetry (DSC) and thermogravimetric analysis (TGA) were performed to assess the thermal properties of the BC samples. Fourier transform infrared spectroscopy (FT-IR) was employed to identify the functional groups present in the BC samples. X-ray diffraction (XRD) was used to determine the crystalline structure of the BC samples. Additionally, positron annihilation lifetime spectroscopy (PALS) and Brunauer–Emmett–Teller (BET) analysis were conducted to measure the free volume and surface area of the BC samples.

## 2. Results and Discussion

### 2.1. Adsorption Kinetics

To examine the underlying mechanism of the adsorption process, pseudo-first-order and pseudo-second-order models are used to test experimental data. The non-linear pseudo-first-order and pseudo-second-order equations can be expressed by Equations (1) and (2), respectively, as follows [22,23]:(1)$$dq_{t}/dt=k_{1}{(q}_{e}-q_{t}),$$(2)$$dq_{t}/dt=k_{2}{\left(q_{e}-q_{t}\right)}^{2},$$
where *q_e_* and *q_t_* (mg g^−1^) are the adsorption capacities at equilibrium and at time *t* (min). *k*_1_ (min^−1^) is the rate constant of the pseudo-first-order model. *k*_2_ (g mg^−1^ min^−1^) is the rate constant of the pseudo-second-order model.

Integrating Equations (1) and (2) and applying the initial conditions, we can obtain the following linear pseudo-first-order and pseudo-second-order equations (Equations (3) and (4)) [22,23]:(3)$$\ln\left(q_{e}-q_{t}\right)=lnq_{e}-k_{1}t,$$(4)$$t/q_{t}=1/(k_{2}q_{e}^{2})+t/q_{e},$$
and the values of *q_e_*, *k*_1_, and *k*_2_ can be calculated from the slopes and the intercepts of the straight lines.

#### 2.1.1. FDBC

The adsorptive behavior of FDBC is illustrated in Figure 1. As shown in Figure 1a, FDBC exhibits a very high adsorption rate at the beginning of the experiment, adsorbing up to 0.5 mg of MB within the first 2 h. This rapid adsorption can be attributed to freeze-drying, which effectively preserves the unique nanoscale porous structure of the pristine BC membranes [21]. Upon immersion in the MB solution, the FDBC undergoes swelling as water molecules penetrate into its porous network. This swelling effect increases the accessible pore volume and facilitates the diffusion of MB molecules into the internal structure. As a result, MB molecules are able to interact with a larger portion of the cellulose nanofibril network, leading to a markedly higher initial adsorption rate.

Additionally, it was observed that an adsorption–desorption equilibrium was attained after 3 days, indicating a relatively slow adsorption process. This slow rate can be explained by the lack of external stimulation, such as shaking, heating, or pH adjustment of the MB solution, allowing adsorption to occur solely through simple Brownian motion of MB molecules. The maximum adsorption capacity (*q_max_*) of FDBC obtained from the experiment is approximately 0.85 mg g^−1^. This value will be compared with the results for ODBC and RDBC in subsequent sections.

Furthermore, as depicted in Figure 1b,c, it is evident that the adsorption data for FDBC fits the pseudo-second-order model better than the pseudo-first-order model. The linear plots of *t*/*q_t_* versus *t* clearly illustrate this, indicating that the kinetics of MB adsorption on FDBC involves a chemical adsorption process through the exchange or sharing of electrons between the adsorbents and adsorbates [24].

#### 2.1.2. ODBC

The adsorptive behavior of ODBC is shown in Figure 2. As illustrated in Figure 2a, unlike FDBC, ODBC does not exhibit a rapid adsorption rate at the beginning of the experiment and takes much longer to reach adsorption equilibrium. We hypothesize that the rapid removal of water content from the BC hydrogel during oven drying induces significant capillary shrinkage and partial pore collapse, which together reduce the accessible free volume of the membrane. Consequently, water and MB molecules require much more time to penetrate the membrane structure and interact with the internal ultrastructure of the cellulose nanofibrils, leading to slower adsorption kinetics compared with FDBC. Interestingly, it is notable that the *q_max_* of ODBC is approximately 1.05 mg g^−1^, which is higher than that of FDBC. This finding may be attributed to the harsher drying environment, which creates a much rougher surface on ODBC compared to FDBC, thereby increasing the surface area available for MB-BC interaction.

Additionally, like FDBC, the adsorption data for ODBC fits the pseudo-second-order model better than the pseudo-first-order model, indicating that the kinetics of MB adsorption on ODBC involves a chemical adsorption process.

#### 2.1.3. RDBC

The adsorptive behavior of RDBC is depicted in Figure 3. As shown in Figure 3a, the adsorption capacity of RDBC falls between that of FDBC and ODBC. Unlike ODBC, RDBC does not experience the harsh conditions of heat-induced water vaporization as ODBC does, nor does it completely avoid pore collapse like FDBC. As a result, RDBC reached adsorption equilibrium after 5 days, slower than FDBC but faster than ODBC. In addition, RDBC does not exhibit a rapid initial adsorption rate like FDBC due to its relatively smooth surface and less porous morphology, which hinder the entry of water and dye molecules. The *q_max_* of RDBC is relatively low, about 0.65 mg g^−1^. This is most likely related to its surface features: compared to ODBC, which exhibits a wrinkled and rough surface that can provide more interaction sites for MB molecules, RDBC presents a smoother surface that limits the number of effective adsorption sites. Moreover, the wrinkled morphology of ODBC may also create localized regions that enhance the accessibility of MB molecules to hydroxyl groups, thereby increasing adsorption efficiency. Such subtle surface morphological differences could have a decisive influence on adsorption performance.

Like FDBC and ODBC, the adsorption kinetics of RDBC fit the pseudo-second-order model better than the pseudo-first-order model, indicating that the MB adsorption process on RDBC involves chemical adsorption.

#### 2.1.4. Summary and Comparisons

The relevant parameters for the kinetic models, calculated from fitting, are summarized in Table 1. It is confirmed the pseudo-second-order model fits the adsorption kinetic data better for all types of dried BC, as indicated by R^2^.

It should be noted that the *q_max_* values of the dried BC samples were extremely low compared with conventional adsorbents such as activated carbon [25]. Several factors contributed to this observation. First, the initial MB concentration used in our tests was only 8 mg L^−1^ with a solution volume of 3 mL, which inherently limited the number of dye molecules available for adsorption. Second, the adsorption tests were carried out under static conditions, without any external stimulation such as heating, stirring, or pH adjustment. Third, the BC membranes were tested in their pristine form, without any chemical modification or compositing. In practical applications, BC is typically modified or incorporated into composites to introduce more active sites and enhance adsorption performance [26]. Under such conditions, and with appropriate external stimulation, the *q_max_* of BC-based materials can be substantially improved.

Furthermore, the adsorption kinetics of FDBC, ODBC, and RDBC are summarized and compared with VWR^®^ filter paper in Figure 4. Within the first three days, the filter paper exhibited a higher overall adsorption capacity than the BC samples. However, during the first day, FDBC demonstrated a faster adsorption rate and higher capacity than the filter paper. This result indicates that, under the same experimental conditions, FDBC possesses significant potential for rapid, short-term adsorption of MB dye.

### 2.2. Adsorption Isotherms

Two equilibrium models were selected to study the adsorption isotherms: Langmuir (Equation (5)) and Freundlich (Equation (6)), as follows:(5)$$q_{e}=q_{m}K_{L}C_{e}/(1+K_{L}C_{e}),$$(6)$$q_{e}=K_{F}C_{e}^{\frac{1}{n}},$$
where *q_e_* is the adsorptive capacity at equilibrium (mg g^−1^) and *C_e_* is the equilibrium concentration (mg L^−1^) in both models, *q_m_* is the maximum adsorption capacity (mg g^−1^), and *K_L_* (L mg^−1^) is the Langmuir isotherm constant. *K_F_* (L g^−1^) is the Freundlich parameter, and *n* is the constant describing the adsorption intensity.

The Equations (5) and (6) can be linearized to Equations (7) and (8), respectively, as follows:(7)$$C_{e}/q_{e}=C_{e}/q_{m}+1/(K_{L}q_{m}),$$(8)$$lnq_{e}=lnK_{F}+(1/n)lnC_{e},$$
and the fitting results are presented in Appendix A, and the correlation parameters of isotherm models are summarized in Table 2.

Based on the results of the linear fitting (Appendix A) and the R^2^ values, although the Langmuir model was applied for comparison, the resulting fits were very poor, as indicated by the extremely low R^2^ values (less than 0.30) and the scattered distribution of data points in the linearized plots (Appendix A). In some cases, forcing the data into the Langmuir equation even produced negative constants, which are physically meaningless and further confirm that this model does not describe the adsorption process of our BC membranes. These inconsistencies reflect the intrinsic inapplicability of the Langmuir assumptions, which are uniform adsorption sites and monolayer coverage, to our heterogeneous BC structures. In contrast, the Freundlich model showed excellent agreement with the experimental data, demonstrating that adsorption on BC is better described as a heterogeneous, multilayer process with non-uniform binding sites. Although the R^2^ value of Freundlich fitting is less than 0.99, it does not significantly impact our ability to use the Freundlich model to explain the adsorptive behavior of our samples, as it is a qualitative rather than a quantitative model. The lower R^2^ value is attributed to the variances among different tested BC samples. Since BC is biosynthesized by bacterial strains and the formation environment is not strictly controlled, obtaining BC membrane samples with consistent structure and properties is challenging.

Building upon the findings of the previous section, the adsorption of MB onto BC is elucidated through the conformity of the experimental data to both the pseudo-second-order kinetic model and the Freundlich isotherm. The pseudo-second-order model suggests that the adsorption process is primarily governed by chemi-sorption, indicating the involvement of chemical interactions such as covalent bonding, ion exchange, or complexation between the dye molecules and the active sites on the BC. The Freundlich isotherm model, which describes adsorption on a heterogeneous surface, implies a distribution of adsorption sites with varying affinities and supports the presence of multilayer adsorption. Furthermore, the cationic nature of MB facilitates electrostatic interactions with the negatively charged functional groups on BC, enhancing adsorption. Hydrogen bonding between the hydroxyl groups of BC and the dye molecules, along with van der Waals forces, also contribute to the adsorption mechanism. These combined mechanisms, chemi-sorption, surface heterogeneity, multilayer adsorption, electrostatic interactions, hydrogen bonding, and van der Waals forces, comprehensively explain the observed adsorption behavior of MB onto BC.

### 2.3. Morphology of Dried BC Membranes

#### 2.3.1. FE-SEM

The FE-SEM images (Figure 5) reveal that FDBC exhibits a more open structure compared to ODBC and RDBC. This observation aligns with our hypothesis that oven and room temperature drying cause capillary shrinkage and pore collapse in BC hydrogels. The diameter of BC fibers ranges from about 50 to 100 nm. However, it should be noted that quantifying BC fiber diameter using FE-SEM images is not precise, because the variations are high on different characterized locations of samples. Additionally, the cross-sectional images for all types of BC clearly show a layered structure, which is attributable to the synthesis process by bacterial strains [27]. This structural characteristic supports the better fit of the Freundlich isotherm model for the adsorption data of BC samples due to their heterogeneous surface.

To gain a deeper understanding of the internal structure of FDBC, a piece of FDBC was mounted on a sample holder and torn apart to expose a plane surface of its inner structure. The FE-SEM results are presented in Figure 6. The images reveal a highly porous structure with numerous voids, which facilitate the rapid adsorption of water and dye molecules at the beginning of our experiments.

#### 2.3.2. CLSM

The CLSM images (Figure 7) demonstrate that FDBC preserves the original mountain-shaped morphology of pristine BC hydrogel. In contrast, ODBC and RDBC lose this morphology and display flatter surfaces with less pronounced projections. Among them, ODBC exhibits a notably rougher surface compared to RDBC, which supports our hypothesis that oven drying induces a wrinkled surface on BC membranes, thereby providing more accessible surface sites for dye adsorption.

#### 2.3.3. AFM

The orientation and morphology of BC nanofibers are depicted in the AFM images (Figure 8). It was found that the AFM technique is not suitable for characterizing one side of FDBC. This is because freeze-drying preserves the original surface structure of the BC hydrogel exceptionally well, resulting in significant roughness, as shown in Figure 7. Additionally, the BC nanofibers are too soft and can be easily displaced by the AFM probe, leading to very fuzzy images, as observed in Figure 8a. Furthermore, it was concluded that dried BC samples exhibit different surface morphologies on each side of the membrane. Notably, for ODBC and RDBC, a flatter surface was observed on one side (Figure 8e,f), indicated by less contrast in the images. This finding suggests that the surface morphology of BC hydrogels can potentially be modified by drying them on different surfaces to optimize adsorption performance.

Overall, the AFM results corroborate the CLSM observations, as both techniques consistently reveal a rougher surface for FDBC and a flatter morphology for ODBC and RDBC. This agreement between the two methods further validates our interpretation of the surface characteristics of dried BC membranes and their influence on adsorption performance.

### 2.4. FT-IR

According to the FT-IR results (Figure 9), the three types of dried BC samples exhibit identical peaks, indicating that the drying methods did not alter their chemical structure. The broad absorption band around 3200–3500 cm^−1^ corresponds to O-H stretching vibrations, indicative of the abundance of hydroxyl groups. Peaks between 2800–3000 cm^−1^ are attributed to C-H stretching from glucose units. Additionally, the peak at approximately 1057 cm^−1^ is characteristic of the asymmetric stretching of the C-O-C bridges, reflective of the β-(1→4) glycosidic linkages. The C-O stretching vibrations, observed between 1020–1100 cm^−1^, correspond to the alcohol groups in the cellulose [28]. Compared to plant-derived cellulose, BC displays variations in peak intensities and positions attributed to its higher purity and crystallinity. The sharpness and intensity of the O-H and C-H stretching peaks, alongside hydrogen bonding influences, are illustrative of the crystalline nature of BC.

### 2.5. XRD

The XRD result is summarized in Figure 10 and Table 3. The crystallinity index (CI, %) was calculated via the PeakFit v4.12 software to perform peak deconvolution. The crystallinity of samples was calculated by dividing the total peak area of all the crystalline peaks at around 14.2°, 16.5°, and 22.55° by the total peak area of all crystalline peaks plus amorphous peak at around 20.5° according to the Equation (9) described in the previous study as follows [29]:(9)$$CI,\%=\frac{\sum A_{cryl}}{\sum A_{cryl}+\sum A_{amph}},$$
where $\sum A_{cryl}$ is the integrated area of all crystalline peaks and $\sum A_{amph}$ is the integrated area of all amorphous peaks.

The original XRD patterns were analyzed using a peak deconvolution method with an assumed Gaussian peak shape. The diffractograms confirmed the material as cellulose I. The diffraction peak at 14.2° corresponds to the (1 0 0) plane of cellulose I_α_ or (1 −1 0) plane of cellulose I_β_. The peak at 16.5° is attributed to the (0 1 0) plane of cellulose I_α_ or (1 1 0) plane of cellulose I_β_. The peak at 22.5° corresponds to the (1 1 0) plane of cellulose I_α_ or (2 0 0) plane of cellulose I_β_. Smaller peaks were disregarded as they do not significantly characterize cellulose I [13,30]. These CI values are consistent with the XRD patterns shown in Figure 10. Specifically, ODBC and RDBC exhibit sharper and more intense diffraction peaks at 14.2°, 16.5°, and 22.5°, along with a weaker amorphous background around 20.5°, which corresponds to their higher CI values (98.56% and 98.14%, respectively). In contrast, FDBC shows broader and less intense crystalline peaks together with a stronger amorphous contribution, consistent with its lower CI value (86.92%). It was found that both oven and room temperature drying methods cause BC nanofibrils to become more densely packed, leading to higher crystallinity. This increased density makes it more difficult for dye molecules to penetrate the OD and RDBC adsorbent, thereby prolonging the time required to achieve adsorption equilibrium.

### 2.6. Thermal Analysis (DSC & TGA)

The thermal analysis (Figure 11) shows that all BC samples exhibit two phases of weight loss. In phase-1 (between 25 °C and 100 °C), the initial weight loss is likely due to moisture removal from the BC. In phase-2 (between 220 °C and 415 °C), significant weight loss is attributed to the depolymerization and pyrolytic decomposition of BC. TGA analysis indicates that ODBC and RDBC remain stable up to 300 °C, after which decomposition begins. In contrast, FDBC starts to decompose at 250 °C. Between 260 °C and 375 °C, approximately 81% of FDBC decomposes, while only about 65% of ODBC and RDBC decompose within a similar temperature range. The first derivative of the TGA curves (Figure 11b,e,h) further confirms that ODBC and RDBC have higher thermal stability compared to FDBC.

The DSC results (Figure 11c,f,i) indicate that all samples undergo an endothermic event upon heating. The glass transition temperatures of ODBC and RDBC are clearly observed, being very similar at approximately 149 °C and 148 °C, respectively. However, a clear glass transition phase is not observed for FDBC, which could be due to its crystalline structure.

### 2.7. BET

The dried BC samples were cryomilled into fine powder via a Restch Cryomill and degassed at 120 °C for 12 h before performing BET. The result is summarized in Appendix A and Table 4. FDBC exhibited the highest BET surface area value, while ODBC and RDBC showed slightly lower values. Although the absolute differences among the samples are small, the consistently higher surface area of FDBC aligns with its enhanced adsorption performance observed in the MB uptake tests. Importantly, FDBC shows a noticeably higher BET surface area compared with the other two samples, whereas the difference between ODBC and RDBC is not statistically significant. The larger surface area of FDBC provides more accessible adsorption sites and facilitates MB diffusion, leading to enhanced adsorption efficiency.

It should be emphasized, however, that the BET measurements were conducted on cryo-milled powders rather than on intact membranes. Cryo-milling inevitably disrupts the intrinsic three-dimensional fibrillar network of BC membranes, especially in FDBC, and thus the obtained BET values may not fully reflect the porosity of the membranes under their actual adsorption state. To overcome this limitation, PALS was further employed to probe the free-volume characteristics of the intact membranes.

### 2.8. PALS

The bulky free volume of BC samples was studied via the PALS technique. The result is summarized in Table 5 and Appendix A.

In Table 5, *τ*_1_, *τ*_2_, *I*_1_, and *I*_2_ represent positron and para-positronium annihilations, which are not the primary focus of this analysis. The parameter *τ*_3_ corresponds to the lifetime of ortho-positronium (o-Ps), and *I*_3_ denotes the corresponding intensities. The last column, *D*, presents the converted pore sizes derived from the o-Ps lifetimes. In this case, larger *D* value represents larger bulky free volume in BC samples. PALS provided additional insight into the microstructural differences among the BC membranes. The larger bulky free volume measured for FDBC indicates that freeze-drying generates a looser fibrillar packing compared to ODBC and RDBC. This structural feature directly explains the faster initial adsorption rate observed for FDBC, since dye molecules can more readily penetrate and diffuse through its network. At the same time, the presence of larger voids also correlates with the lower thermal stability revealed by TGA, reflecting a trade-off between adsorption accessibility and structural robustness. These results demonstrate how PALS measurements support the adsorption kinetics and stability trends, integrating nanoscale free-volume information into the overall adsorption mechanism.

### 2.9. Summary of Characterization Tests

The characterization results demonstrate that FDBC possesses distinct structural features compared with ODBC and RDBC. Specifically, FDBC exhibits the highest BET surface area and bulky free volume together with lower crystallinity, which collectively contribute to its rapid adsorption of dye molecules. However, this advantage comes at the expense of weaker thermal stability relative to ODBC and RDBC. Surface morphology analysis further indicates that ODBC has a much rougher surface than RDBC, consistent with its better adsorption performance. Cross-sectional FE-SEM images confirm that all dried BC membranes exhibit a layered architecture, in line with the assumptions of the Freundlich isotherm. In addition, 3D structural data of FDBC obtained by XRM are presented in Appendix A. These images provide a rare visualization of the internal architecture of FDBC and further support its highly porous and open structure. We believe this information will be useful for future investigations into the structure–property relationships of BC-based materials.

## 3. Conclusions

This study compared the adsorption of MB on FDBC, ODBC, and RDBC membranes. FDBC exhibited the fastest initial adsorption rate owing to its preserved porous structure, whereas ODBC achieved the highest overall adsorption capacity. The adsorption behavior fitted well with the pseudo-second-order kinetic model and the Freundlich isotherm, suggesting that the process is dominated by chemisorption and multilayer adsorption on heterogeneous surfaces. The key mechanisms involve electrostatic interactions between cationic MB and hydroxyl groups of BC, as well as hydrogen bonding and van der Waals forces. Together, these results demonstrate that drying-induced morphological differences and surface functionalities jointly govern the adsorption performance of BC membranes.

## 4. Materials and Methods

### 4.1. Materials

Amazon Brand black tea, Domino granulated white sugar, Lucy’s Family Owned—raw apple cider vinegar, Clorox 30966 Concentrated Regular Bleach, and kombucha SCOBY were purchased from Amazon, Seattle, WA, USA and used as received without further modification. VWR^®^ Qualitative Filter Paper, Grade 413 was purchased from VWR, Radnor, PA, USA and used as received without further modification. Methylene Blue, hydrogen chloride, and potassium hydroxide solution 1N (N/10) (Certified) were purchased from Sigma-Aldrich, St. Louis, MO, USA, and used as received without further modification.

### 4.2. Bacterial Cellulose Biosynthesis

Tea was brewed by using Amazon Brand black tea (4.5 g, two tea bags) and 750 mL of boiling deionized (DI) water and allowed to stand for 20 min. The tea bags were then removed, granulated white sugar (150 g) was dissolved in the tea, and 150 mL raw apple cider vinegar was added to the tea. The culture medium was then allowed to cool to room temperature and kombucha SCOBY was added. The sweetened tea with the starting SCOBY was then covered with a cotton cloth and allowed to stand in the incubator for 21 days. Then the new kombucha pellicle formed at the air–water interface was carefully removed, washed with DI water and pat dried with Kimwipes^®^ and used for the purification procedure.

### 4.3. Bacterial Cellulose Purification

To purify the BC hydrogels, the BC membranes were soaked in a 1 M potassium hydroxide solution at room temperature for 48 h to remove non-cellulosic materials such as proteins and nucleic acids from bacterial cells and the culture broth. After most of the media was removed, the BC membranes were transferred to a 0.5 M acetic acid bath to neutralize the base. Following 1 h of neutralization, the cellulose was repeatedly rinsed with DI water until it reached a neutral pH. In the subsequent step, the membrane was fully immersed in bleach for at least 20 min until it became completely white, removing any residual coloration. Finally, the membrane was stored in fresh DI water at 4 °C for further experiments.

### 4.4. Preparation of Dried/Rehydrated Dried BC Samples

FDBC samples were prepared using a LABCONCO FreeZone 2.5 L −50 °C benchtop freeze dryer. The samples were dried at −50 °C and a pressure of approximately 10 Pa for at least 48 h until fully dried. For ODBC, the purified BC membranes were placed on sterilized Petri dishes and dried in an oven at 60 °C for a minimum of 3 h until completely dry. RDBC samples were produced by placing the purified BC membranes on sterilized Petri dishes and leaving them in a laboratory fume hood at room temperature (around 20 °C) for at least 48 h until thoroughly dried. To obtain rehydrated BC samples, the dried BC samples were immersed in DI water at room temperature for a minimum of 72 h. Subsequently, the rehydrated (RH) BC samples were dried again using the same method as the initial drying process to obtain rehydrated FDBC, ODBC, and RDBC samples, respectively.

### 4.5. The UV-Vis Test: Methylene Blue Solution Adsorption Kinetics Study

As shown in Appendix A, a piece of cut FD/OD/RDBC membrane or filter paper membrane (dry mass about 0.016 g) was placed at the bottom of a 1 cm × 1 cm cuvette, and the cuvette was filled with 3 mL 8 mg L^−1^ MB solution. The top of the cuvette was sealed with Parafilm to prevent evaporation of the solution during the experiment. The cuvette was then placed in the Cary 60 UV-Vis spectrophotometer for testing. Absorbance readings of the sample were recorded at discrete time intervals at a wavelength of 663 nm and converted to corresponding concentrations using the Beer–Lambert law (Equation (10)) as follows [20]:(10)A=ϵbc,
where *A* is the absorbance of the solution, ϵ is absorptivity (L mg^−1^ cm^−1^), *b* is the length of light path, which is 1.0 cm in our case, and *c* is the concentration of solution (mg L^−1^). The amount of MB adsorbed from the solution was then calculated by the following equation (Equation (11)):(11)$$q_{e}=\left[\left(C_{0}-C_{e}\right)V\right]/m,$$
where *q_e_* (mg g^−1^) is the adsorption capacity at equilibrium, *C*_0_ and *C_e_* (mg L^−1^) are the initial and equilibrium concentrations, and *V* (L) and *m* (g) are the volume of the solution and the weight of the BC used.

### 4.6. Characterization Tests Method

The surface morphology of dried BC membranes was characterized using SU8700 FE-SEM (Hitachi, Japan), Verios 460L FE-SEM (FEI, Hillsboro, OR, USA), VKx1100 CLSM (Keyence, Japan), and Asylum MFP-3D Classic AFM (Oxford Instruments, Abingdon, UK). Prior to FE-SEM analysis, samples were coated with Au powder to enhance conductivity. Cross-sections of the samples were prepared using a UC7 Cryo Ultramicrotome (Leica, Wetzlar, Germany) and examined with a Bio-TEM HT7800 (Hitachi, Japan). The 3D structure of the membranes was observed using an Xradia 510 Versa 3D X-ray Tomography System (Zeiss, Oberkochen, Germany) at 50 kV with a 4× objective lens. High-resolution images, with a pixel size of 1.87 μm, were recorded in 8-bit TIFF format using Scout and Scan Control System Reconstructor-16.2 software. These reconstructed TIFF images were then imported into Dragonfly 3D World: Version 2024.1 software (Object Research Systems, Montréal, QC, Canada) for advanced analysis of both 2D and 3D data, with adjustments made to window leveling, contrast, and intensity (brightness) for optimal visualization.

Thermal performance was evaluated using a Discovery DSC 250 (TA Instruments, New Castle, DE, USA) with an RCS cooler, over a temperature range of 25 °C to 300 °C at a heating rate of 10 °C min^−1^, and a Discovery TGA 550 (TA Instruments, New Castle, DE, USA) over a temperature range of 0 °C to 650 °C at a heating rate of 10 °C min^−1^. Functional groups were identified using an iS50 FT-IR (Thermo Fisher, Waltham, MA, USA). The crystalline structure of the samples was analyzed using a SmartLab XRD (Rigaku, Tokyo, Japan) operating at 40 kV and 44 mA with a CuKa source, with a scan step size of 0.02 degrees, a dwell time of 0.5 s per step, and a scan range from 3.0 to 30.0 degrees.

PALS tests were conducted at room temperature using a bulk spectrometer at the NC State PULSTAR Reactor Laboratory. For ODBC and RDBC membrane samples, smaller pieces were cut and stacked to create 8–12 layers on each side of a Na-22 positron source, forming a sample–source–sample sandwich. The FDBC sample, with a thickness exceeding 1 mm, was only cut but not stacked. The assembly was wrapped in aluminum foil and placed between two photomultiplier tubes (PMTs) to record the start and stop signals of the positrons. Each measurement recorded approximately 6 million annihilation events.

Dried BC samples were cryo-milled using a CryoMill (Retsch, Haan, Germany) at −196 °C for 10 min at a frequency of 25 Hz before conducting BET tests. Approximately 100 mg of each BC powder sample was degassed using a VacPrep 061 Sample Degas System (Micromeritics, Norcross, GA, USA) at 120 °C for 12 h to thoroughly remove ambient water and gas. BET surface area analysis was then performed using a TriStar II PLUS (Micromeritics, Norcross, GA, USA) at −196 °C with N_2_ as the adsorbent gas.

## Figures and Tables

**Figure 1 gels-11-00721-f001:**
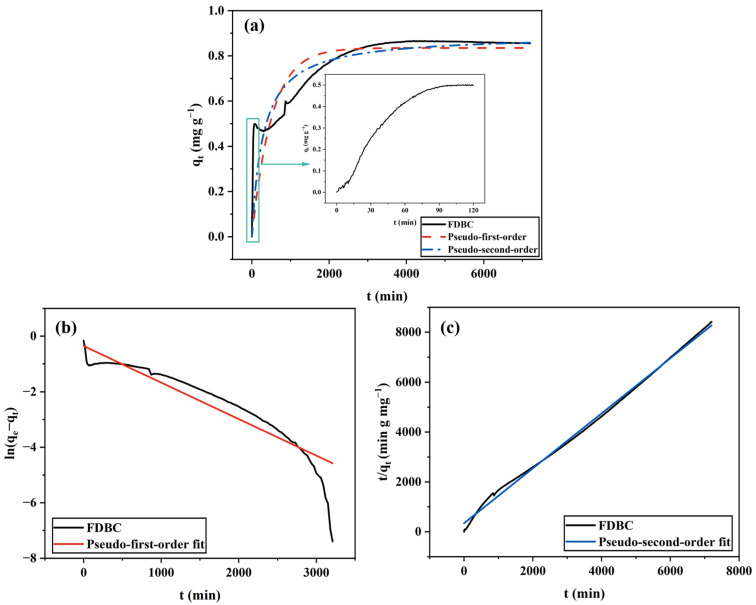
FDBC adsorption kinetics fitted by (**a**) non-linear pseudo-first-order and pseudo-second-order models. The inset of (**a**) shows the detailed adsorptive behavior of FDBC during the first 2 h after the experiment began; (**b**) linear pseudo-first-order model; (**c**) linear pseudo-second-order model.

**Figure 2 gels-11-00721-f002:**
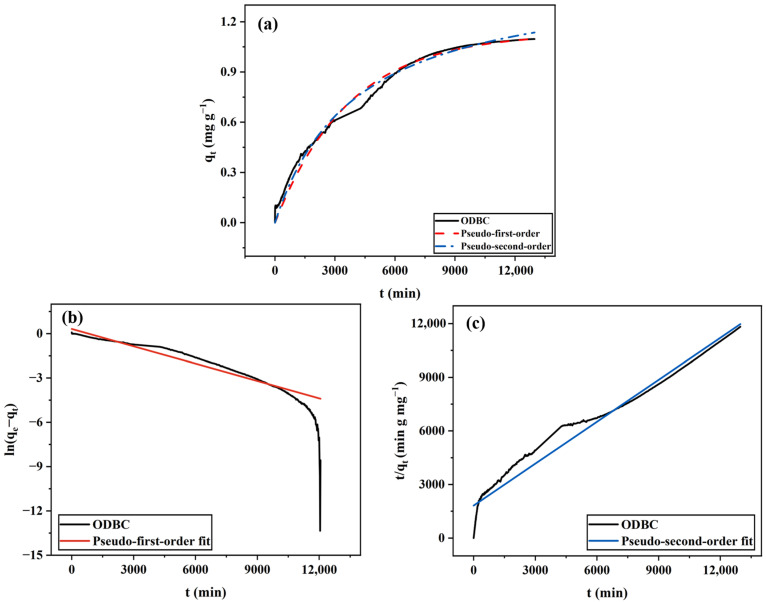
ODBC adsorption kinetics fitted by (**a**) non-linear pseudo-first-order and pseudo-second-order models; (**b**) linear pseudo-first-order model; (**c**) linear pseudo-second-order model.

**Figure 3 gels-11-00721-f003:**
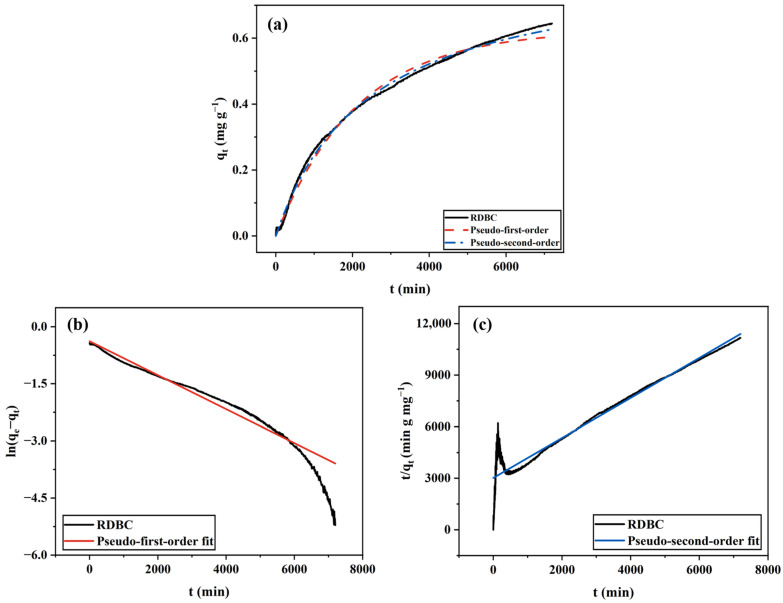
RDBC adsorption kinetics fitted by (**a**) non-linear pseudo-first-order and pseudo-second-order models; (**b**) linear pseudo-first-order model; (**c**) linear pseudo-second-order model.

**Figure 4 gels-11-00721-f004:**
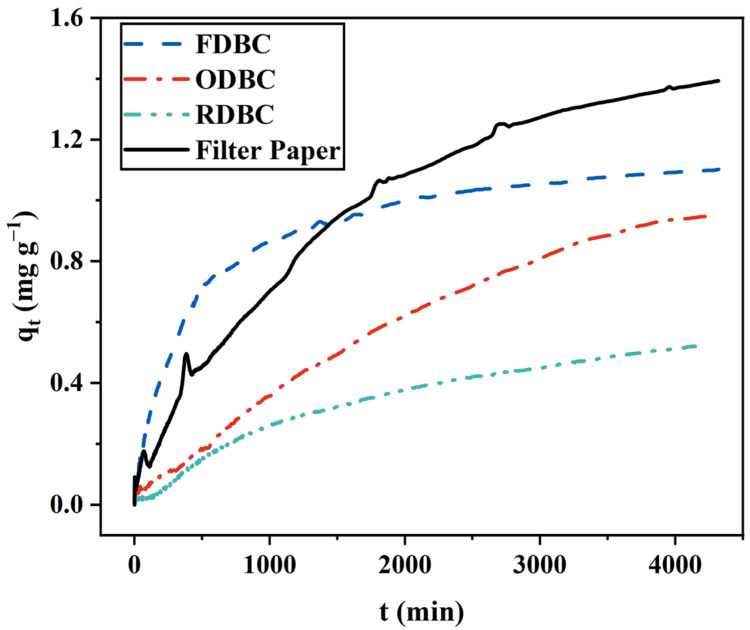
Adsorption behavior of dried BC membranes compared with VWR^®^ filter paper.

**Figure 5 gels-11-00721-f005:**
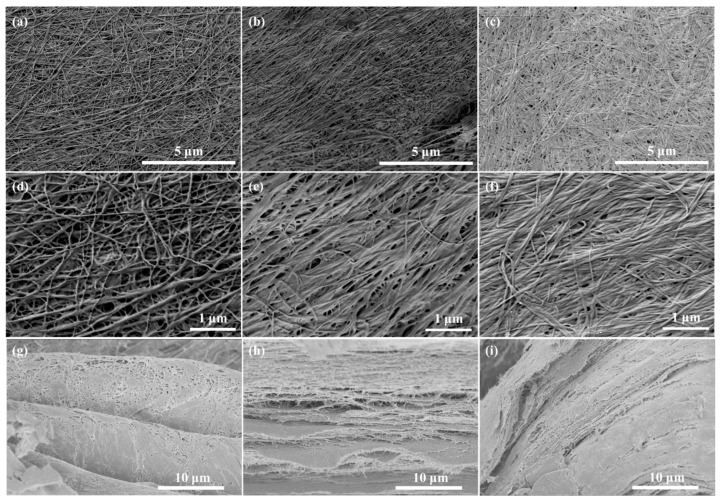
FE-SEM images of (**a**,**d**,**g**) FDBC, (**b**,**e**,**h**) ODBC, and (**c**,**f**,**i**) RDBC, with (**a**–**f**) showing plane views and (**g**–**i**) showing cross-sections, taken at (**a**–**c**) 10,000× magnification, (**d**–**f**) 25,000× magnification, and (**g**–**i**) 3500× magnification.

**Figure 6 gels-11-00721-f006:**
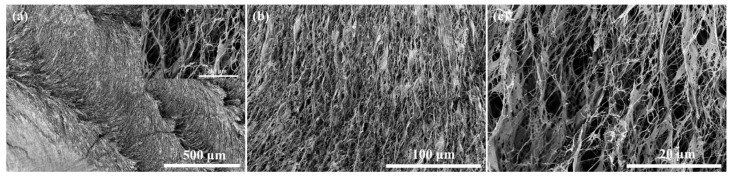
FE-SEM images of the internal structure of FDBC: (**a**) 80× magnification with an inset at 5000× magnification; (**b**) 500× magnification; (**c**) 2500× magnification.

**Figure 7 gels-11-00721-f007:**
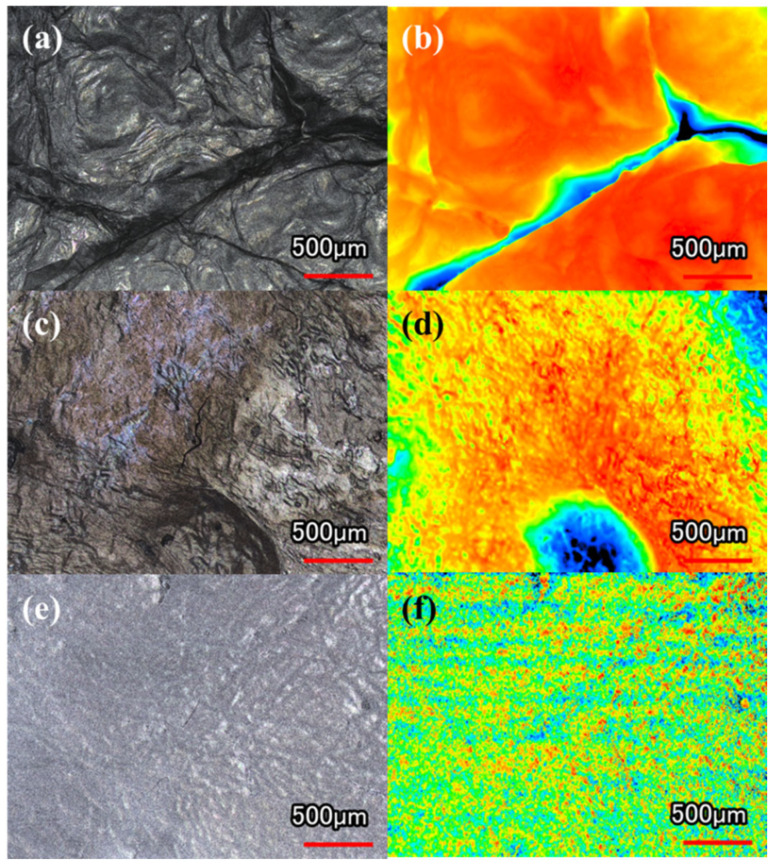
CLSM images of (**a**,**b**) FDBC, (**c**,**d**) ODBC, and (**e**,**f**) RDBC; (**a**,**c**,**e**) display laser and optical views, while (**b**,**d**,**f**) present height information images.

**Figure 8 gels-11-00721-f008:**
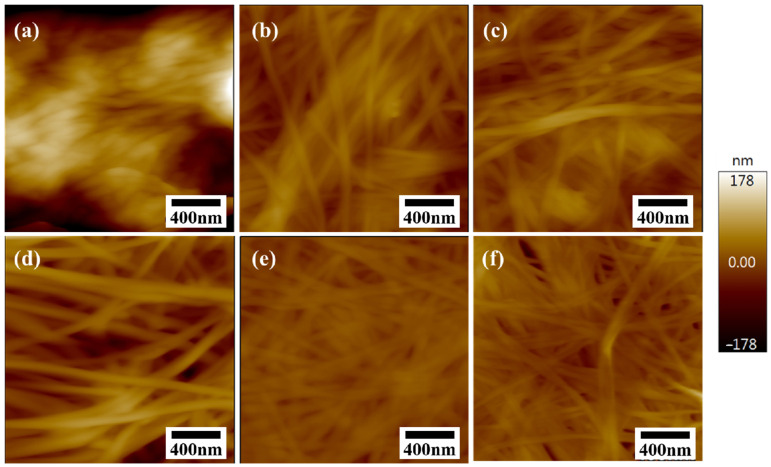
AFM images of both sides of (**a**,**d**) FDBC, (**b**,**e**) ODBC, and (**c**,**f**) RDBC; (**a**–**c**) depict the surfaces exposed to air during the drying process, while (**d**–**f**) depict the surfaces that were in contact with the Petri dish or vacuum bottle.

**Figure 9 gels-11-00721-f009:**
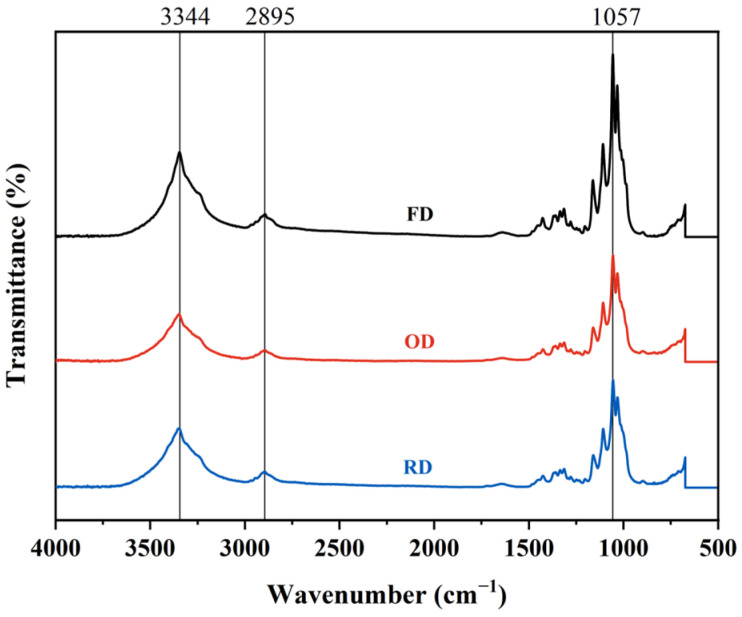
FT-IR analysis of dried BC membrane samples.

**Figure 10 gels-11-00721-f010:**
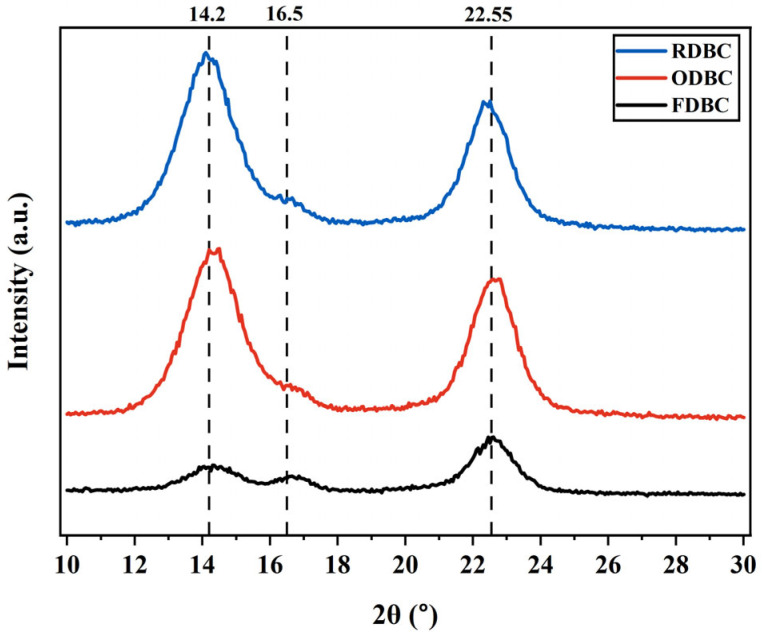
XRD pattern of dried BC membrane samples.

**Figure 11 gels-11-00721-f011:**
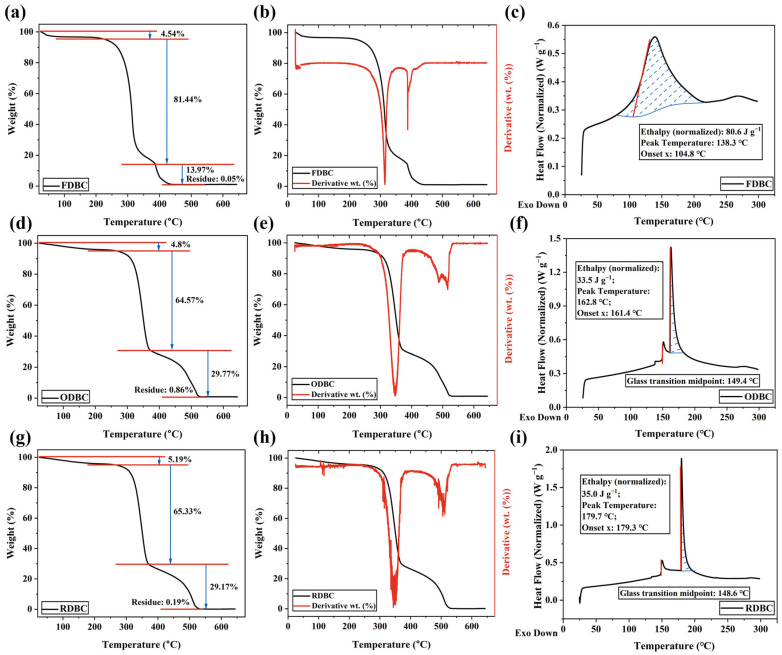
Thermal analysis of (**a**–**c**) FDBC, (**d**–**f**) ODBC, and (**g**–**i**) RDBC are provided, with (**a**,**b**,**d**,**e**,**g**,**h**) showing TGA results and (**c**,**f**,**i**) showing DSC results. The blue lines in (**a,d,g**) and the shaded areas in (**c**,**f**,**i**) indicate auxiliary marks for weight-loss regions in TGA and enthalpy calculation ranges in DSC, respectively, provided to facilitate interpretation.

**Table 1 gels-11-00721-t001:** Pseudo-first and pseudo-second-order constants and R^2^ values for the adsorption of MB onto different types of BC membranes.

BC Type	Pseudo-First-Order Model			Pseudo-Second-Order Model		
	$k_{1}\;({min}^{-1})$	$q_{e\;}(mg\;g^{-1})$	$R_{1}^{2}$	$k_{2}\;(g\;{{mg}^{-1\;}min}^{-1})$	$q_{e}\;(mg\;g^{-1})$	$R_{2}^{2}$
FDBC	0.0020	0.84	0.726	0.0038	0.89	0.829
ODBC	0.0002	0.91	0.995	0.0001	1.30	0.997
RDBC	0.0005	0.62	0.994	0.0005	0.84	0.998

**Table 2 gels-11-00721-t002:** Isotherm constants and R^2^ values for the adsorption of MB onto different types of BC membranes.

BC Type	Langmuir Parameter			Freundlich Parameter		
	$K_{L\;}(L\;{mg}^{-1})$	$q_{m\;}(mg\;g^{-1})$	$R^{2}$	$K_{F\;}(L\;g^{-1})$	n	$R^{2}$
FDBC	0.067	14.25	0.276	0.868	1.01	0.954
ODBC	−0.074	−25.77	0.047	2.172	0.94	0.946
RDBC	−0.036	−20.75	0.210	0.783	0.93	0.986

**Table 3 gels-11-00721-t003:** Crystallinity index of BC samples.

BC Type	CI (%)
FDBC	86.92
ODBC	98.56
RDBC	98.14

**Table 4 gels-11-00721-t004:** BET surface area of BC samples.

BC Type	BET Surface Area (m^2^ g^−1^)
FDBC	5.31 ± 0.04
ODBC	3.60 ± 0.03
RDBC	3.25 ± 0.05

**Table 5 gels-11-00721-t005:** The fitted positron lifetimes and intensities from 3-lifetime free fittings.

BC Type	$\tau_{1}$ (ps)	$I_{1}$ (%)	$\tau_{2}$ (ps)	$I_{2}$ (%)	$\tau_{3}$ (ns)	$I_{3}$ (%)	D (nm)
FDBC	208	46	452 ± 1	44.4 ± 0.3	2.53 ± 0.01	9.7 ± 0.2	0.661 ± 0.001
ODBC	267	52	535 ± 3	38.9 ± 0.8	2.04 ± 0.01	9.5 ± 0.5	0.579 ± 0.001
RDBC	274	54	558 ± 9	36 ± 2	2.19 ± 0.01	10.0 ± 1.2	0.606 ± 0.001

## Data Availability

Dataset available on request from the authors.

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
