# Peer review of "Adsorption Characteristics of Bacterial Cellulose Membranes Toward Methylene Blue Dye in Aqueous Environment"

_gels, 2025, doi:10.3390/gels11090721_

Round 1
Reviewer 1 Report
Comments and Suggestions for Authors
Authors have submitted a manuscript titled "Adsorption Characteristics of Bacterial Cellulose Membranes Toward Methylene Blue Dye in Aqueous Environment." While the work presents some interesting findings, it lacks novelty in certain areas and requires significant revisions. Despite this, the manuscript can be considered for publication after addressing the comments.

Reviewer 2 Report
Comments and Suggestions for Authors
This manuscript investigates how different drying methods (freeze-drying, oven-drying, and room-temperature drying) influence the structural characteristics and adsorption behavior of bacterial cellulose (BC) toward methylene blue (MB). The study is well-structured, presents thorough characterization (FE-SEM, CLSM, AFM, FTIR, XRD, DSC/TGA, BET, PALS), and combines adsorption kinetics and isotherm analyses. The work provides useful insights for tailoring BC-based adsorbents in water purification. However, several issues limit its impact, novelty, and clarity.
Major Weaknesses
- Novelty is limited – Previous studies have already explored the effect of drying methods on BC structure and adsorption (e.g., Zeng et al., 2014; Marchiori et al., 2024). This work largely confirms known phenomena (freeze-drying preserves porosity; oven-drying increases roughness but collapses pores). The manuscript needs to better highlight what new insight it brings.
- Low adsorption capacities – The reported qmax values (0.65–1.05 mg/g) are extremely low compared to many adsorbents (activated carbons, modified cellulose, composites often reach 100–500 mg/g for MB). This raises questions about the practical significance of BC in its current form. The manuscript should discuss why the capacities are so low and how they could be improved.
- Experimental conditions are simplistic – The adsorption tests are static (no stirring, no pH variation, low initial concentration). This may not reflect real wastewater conditions. The lack of control experiments (e.g., effect of pH, ionic strength, competing ions) weakens environmental relevance.
- Inconsistencies in modeling – The Langmuir model fitting yielded negative constants (Table 2), which is physically meaningless. This indicates poor experimental design or insufficient concentration ranges. These results should be addressed explicitly rather than presented without critique.
- Overuse of advanced characterization without integration – Techniques like PALS and XRM are applied, but the discussion does not convincingly integrate them into adsorption behavior explanations. It reads more like a “characterization catalog” than a coherent mechanistic study.
I suggest a Major Revision, as that the authors need to better justify novelty, contextualize adsorption performance, and streamline the manuscript before it can be considered for publication.
Reviewer 3 Report
Comments and Suggestions for Authors
Attached.

Round 2
Reviewer 1 Report
Comments and Suggestions for Authors
The authors have addressed all comments from the reviewers and made significant enhancements to the manuscript. With these improvements in mind, I recommend accepting this revised manuscript for publication.
Author Response
Dear Reviewer,
We sincerely thank you for your positive feedback and recommendation for acceptance. We truly appreciate your constructive comments and detailed suggestions throughout the review process, which have greatly improved the quality and clarity of our manuscript.
Thank you again for your kind support.
Sincerely,
Zimu Hu
Reviewer 2 Report
Comments and Suggestions for Authors
A very recent study on bacterial cellulose interpenetrating network may be a helpful reference for your work: doi: 10.1038/s41598-024-56534-z
Author Response
We sincerely thank the reviewer for suggesting this recent reference (doi: 10.1038/s41598-024-56534-z). After carefully reviewing it, we have incorporated the citation into the Introduction section, where we discuss the advantages of bacterial cellulose hydrogels over plant-derived cellulose hydrogels. We also replaced one older reference with this recent work to ensure our manuscript reflects the most up-to-date progress in the field. We believe this addition further strengthens the relevance and timeliness of our study.
Reviewer 3 Report
Comments and Suggestions for Authors
The manuscript has undergone substantial improvement; however, some comments still require attention, as noted below.
Comment 7-4: In my opinion, the lower average pore diameter of FDBC is in accordance with its higher specific surface area. That means that the sample appear to have more pores of lower diameter, leading to a higher surface area. Probably, there is no contradiction with FE-SEM. For this reason, images at higher magnifications were needed.
Comment 11-4: The morphological distinction between ODBC and RDBC (rough or smooth) may not be enough and raise concerns to explain the different adsorption behaviour despite their similar bulk characteristics.
